# Transmission of tuberculosis between foreign-born and Finnish-born populations in Finland, 2014–2017

Pirre Emilia Räisänen[1,2]*, Marjo Haanperä[3], Hanna Soini[2], Petri Ruutu[2], J. Pekka Nuorti[1,2], Outi Lyytikäinen[2]

1 Health Sciences unit, Faculty of Social Sciences, Tampere University, Tampere, Finland, 2 Infectious Disease Control and Vaccinations Unit, Department of Health Security, Finnish Institute for Health and Welfare, Helsinki, Finland, 3 Expert Microbiology Unit, Department of Health Security, Finnish Institute for Health and Welfare, Helsinki, Finland

☯ These authors contributed equally to this work.
* pirre.raisanen@thl.fi

**Data Availability Statement:** The datasets generated and/or analyzed during the current study are not publicly available due to the possibility of identification of a case. The authors do not have

## Abstract

We describe the epidemiology of tuberculosis (TB) and characterized Mycobacterium tuberculosis *(M. tuberculosis)* isolates to evaluate transmission between foreign-born and Finnish-born populations. Data on TB cases were obtained from the National Infectious Disease Register and denominator data on legal residents and their country of birth from the Population Information System. *M. tuberculosis* isolates were genotyped by spoligotyping and Mycobacterial Interspersed Repetitive Unit Variable Number Tandem Repeat (MIRU-VNTR). We characterized clusters by age, sex, origin and region of living which included both foreign-born cases and those born in Finland. During 2014–2017, 1015 TB cases were notified; 814 were confirmed by culture. The proportion of foreign-born cases increased from 33.3% to 39.0%. Foreign-born TB cases were younger (median age, 28 vs. 75 years), and had extrapulmonary TB or multidrug-TB more often than Finnish-born cases (P<0.01 for all comparisons). Foreign-born cases were born in 60 different countries; most commonly in Somalia (25.5%). Altogether 795 isolates were genotyped; 31.2% belonged to 80 different clusters (range, 2–13 cases/cluster). Fourteen (17.5%) clusters included isolates from both Finnish-born and foreign-born cases. An epidemiological link between cases was identified by (epidemiological) background information in two clusters. Although the proportion of foreign-born TB cases was considerable, our data suggests that transmission of TB between foreign and Finnish born population is uncommon.

## Introduction

Tuberculosis (TB) is a worldwide disease, and the most affected areas are the South-East Asia, Western Pacific regions, and Africa [1]. Increasing migration from high TB incidence countries is a challenge for national TB programmes in low TB incidence countries (<10/100 000 population), such as Finland. Our previous nationwide, population-based study showed that

permission to share the data. Qualified researchers can apply for access to the data through the Health and Social Data Permit Authority (Findata) (https://www.findata.fi/en/ or https://www.findata.fi/en/services/data-permits/) and by following the protocol outlined in the Methods section.

**Funding:** PE Räisänen was supported by the Foundation of the Finnish Anti-Tuberculosis Association (https://www.tb-foundation.org/), Tuberculosis Foundation of Tampere (http://www.tuberkuloosisaatio.fi/) and The Research Foundation of Pulmonary Diseases (https://www.hengitysliitto.fi/en/organisation-respiratory-health/funding-and-partners/research-foundation-pulmonary-diseases). The funders had no role in study design, data collection and analysis, decision to publish, or preparation of the manuscript.

**Competing interests:** The authors have declared that no competing interests exist.

during 1995–2013 the number of notified TB cases in foreign-born persons tripled as the foreign-born population from high-incidence countries grew [2]). The proportion of all TB cases who were foreign-born increased more than fivefold reflecting both increased migration and the decrease in the number of Finnish-born TB cases in Finland. Somali-born population represented over 30% of the foreign-born TB cases while Somalis were the fourth largest group of migrants in Finland [2]. In 2015–2016, when the large influx of asylum seekers reached Europe, Finland received approximately 35,000 asylum seekers and the majority of them were born in Iraq, Afghanistan, Somalia or Syria [3].

Genotyping has provided important insights into the molecular epidemiology of TB in many low-incidence countries [4–6]. The risk of transmission from migrant to native populations is generally considered low [7, 8]. In low-incidence countries, TB transmission mainly occurs within households [9–11]. Studies suggest that the risk of transmission is increased within migrant households and migrant communities, but not in host populations. A previous study conducted in Finland in 2013 showed that only 10% of clusters included both foreign- and Finnish-born cases [12].

In this study, we describe the epidemiology of TB in Finland during 2014–2017 and characterize Mycobacterium tuberculosis *(M. tuberculosis)* isolates by genotyping methods to evaluate transmission between foreign- and Finnish-born populations.

## Materials and methods

### National Infectious Disease Register

In Finland, clinical microbiology laboratories notify new *M. tuberculosis* isolations directly to the National Infectious Disease Register (NIDR), maintained by the Finnish Institute for Health and Welfare (THL), and submit isolates to the Mycobacterial Reference Laboratory at THL for drug susceptibility testing and genotyping. In addition, physicians notify clinically confirmed TB cases to NIDR; reporting is mandatory. All cases notified to NIDR during 2014–2017 were included in the study. The TB surveillance case definition includes all cases confirmed by culture, sputum smear, nucleic acid amplification and/or histology, as well as clinically diagnosed TB, if a decision to give full TB treatment is made [13]. Each notification includes a unique national identifier if available, name, date of birth, gender, and the notification of the physician includes in addition country of birth, nationality, place of residence and treatment, dates of symptom onset and diagnosis, diagnostic method, clinical presentation (pulmonary/extrapulmonary TB), and additional information on the contact tracing (free text column on the notification form). A foreign-born in this study is defined as a person whose country of birth is not Finland and if the data is not available the most recent nationality is not Finnish.

### Laboratory methods and identification of clusters

Culture-positive *M. tuberculosis* isolates are routinely sent to the THL Mycobacterial Reference Laboratory for drug susceptibility testing and genotyping by spoligotyping and Mycobacterial Interspersed Repetitive Unit Variable Number Tandem Repeat (MIRU-VNTR) (24 loci) according to standard protocols [14, 15]. DNA was extracted from bacterial colonies grown on solid medium using the CTAB (cetyl trimethylammonium bromide) method [16, 17]. Spoligotyping was performed using spoligotyping membranes obtained from Ocinum Biosolutions (Hyderabad, P.A., India) or produced in-house as previously described by Kamerbeek *et al.* [15]. The MIRU-VNTR analysis was performed at GenoScreen (Lille, France) using the MIRU-VNTR Kits of the same company.

The resulting spoligotype patterns were compared to the data in the international SITVIT-WEB 2 database [18]. An isolate was assigned a shared type (SIT) and a lineage if the same spoligotype was found in the database. The MIRU-VNTR results were analysed by the MIRU-VNTR plus database [19] using the Bionumerics 6.6 software (Applied Maths, Sint-Martens-Latem, Belgium). When two or more *M. tuberculosis* isolates matched by genotyping methods (i.e., identical spoligotype and MIRU-VNTR patterns), they were considered a genotype cluster. Clustering of isolates with incomplete MIRU-VNTR profile due to missing/undetectable, double, or non-numerical (e.g. 3s) results was performed with help of UPGMA (unweighted pair group method with arithmetic mean) tree generated in the Bionumerics software. Clusters which included Finnish- and foreign-born cases were evaluated by using the additional information on contact tracing: type of social contacts such as schoolmates, friends, relatives, and family members. The genotypes of the mixed clusters were compared with a local genotyping database obtained from *M. tuberculosis* strains isolated in Finland to determine whether the clusters had been detected in Finland earlier.

## Population data

The number of people residing legally in Finland by year and country of birth was obtained from the National Population Information System [20].

## Data analysis and statistics

Statistical significance for categorical variables was analysed with the $\chi 2$ test or Fisher's exact test. Continuous variables were analysed using the Mann–Whitney U test. P-value of $<0.05$ was considered statistically significant.

The annual incidence rate was calculated by the annual number of cases divided by population at the end of the year. A Poisson regression model was used to assess the log-linear trend in annual incidence rates.

IBM SPSS v. 25.0 (SPSS Inc., USA) and Microsoft Excel (Microsoft Corp., USA) were used to analyse the data.

Ethical approval was not applicable as data in this study were analysed within the epidemiological research purposes authorized by the Finnish Communicable Diseases Act 1227/2016, 42 §. Therefore, ethical approval was deemed unnecessary.

## Results

From 2014 to 2017, a total of 1015 TB cases were identified in the NIDR; 814 were confirmed by culture. Of the cases, 579 (57.0%) were Finnish-born, 415 (40.9%) were foreign-born and 21 (2.1%) had no information on country of birth or nationality. Of the 415 foreign-born TB cases, 25.5% were born in Somalia, 7.0% in Afghanistan, 6.3% in Thailand and 5.3% in Vietnam; the remaining 232 cases were born in 56 different countries. Foreign-born TB cases were significantly younger compared to Finnish-born cases (P<0.01) (Table 1). Extrapulmonary TB and multidrug-resistant TB (MDR-TB) were more common among foreign-born cases (p<0.01).

The annual incidence of TB decreased from 4.8/100 000 population in 2014 to 4.5/100 000 population in 2017, being highest 5.0/100 000 population in 2015, but the decreasing trend was not statistically significant (p = 0.16) (Table 2). However, there was a significant decreasing trend among Finnish-born population (average annual decrease, 8.7%; p<0.01). The number and the incidence of foreign-born TB cases increased from 2014 to 2015 but fluctuated thereafter during 2015–2017 (p = 0.72). The proportion of foreign-born cases was highest (47.6%) in 2016.

**Table 1. Characteristics of Finnish-born and foreign-born TB cases in Finland, 2014–2017.**

| | | Finnish-born | Foreign-born | *P*-value |
|---|---|---|---|---|
| | | (n = 579) | (n = 415) | |
| Median age, years (range) | | 75 (0–101) | 28 (2–87) | <0.01 |
| Age group, years n (%) | | | | |
| | 0–14 | 13 (2.2) | 19 (4.6) | |
| | 15–29 | 17 (2.9) | 203 (48.9) | |
| | 30–44 | 28 (4.8) | 119 (28.7) | |
| | 45–59 | 62 (10.7) | 49 (11.8) | |
| | 60–74 | 167 (28.8) | 14 (3.4) | |
| | 75- | 292 (50.4) | 11 (2.7) | |
| Male cases, n (%) | | 346 (59.8) | 235 (57.8) | 0.543 |
| Pulmonary TB, n (%) | | 448 (77.4) | 288 (72.1) | <0.01 |
| | Smear positive, n (%) | 150 (33.5) | 107 (37.2) | 0.02 |
| Extrapulmonary TB, n (%) | | 131 (22.6) | 127 (30.6) | <0.01 |
| MDR-TB, n (%) | | 5 (0.9) | 21 (5.3) | <0.01 |

A total of 795 isolates from culture-positive cases were characterised by spoligotyping and MIRU-VNTR. Of the isolates, 248/795 (31.2%) belonged to 80 different clusters (range, 2–13 isolates/cluster): 42 (52.5%) clusters included isolates only from Finnish-born cases and 24 (30%) only from foreign-born cases, 13 of these having only cases born in the same country within a cluster. A total of 14 (17.5%) clusters included Finnish- and foreign-born cases, i.e. were mixed clusters, consisting of 9.6% (76/795) of the cases. Overall, 25.0% and 24.1% of the Finnish-born and foreign-born-cases were clustered, respectively.

Based on the spoligotyping results, 7 different lineages were detected among the mixed clusters. Beijing, Latin American & Mediterranean (LAM) and T lineages were the most common, including 5, 3 and 3 clusters, respectively. Altogether 81.6% isolates belonged to these most common lineages.

Altogether 76 isolates belonged to the 14 mixed clusters: 39 (51.3%) were from Finnish-born cases, 36 (47.4%) from foreign-born, and one of unknown origin. Foreign-born cases originated from Europe (9 cases), Asia (13 cases) and Africa (14 cases, 6/14 from Somalia) (Table 3). The majority of the isolates, 53 (70%) were from male cases and the median age of the cases was 42 years (range, 13–86). Five cases (6.6%) were under the age of 18. Out of the 76 isolates, 69 (91%) were from pulmonary TB cases, and 38 (55%) of these were smear positive.

**Table 2. Incidence of TB cases in the whole, Finnish-born and foreign-born populations in Finland, 2014–2017.**

| | All cases | | | Finnish-born cases | | | Foreign-born cases | | |
|---|---|---|---|---|---|---|---|---|---|
| Year | Population in Finland | No. of cases* | Incidence (95%CI) | Finnish-born population in Finland n (%) | No. of cases (%) | Incidence (95%CI) | Foreign-born population in Finland n (%) | No. of cases (%) | Incidence (95%CI) |
| 2014 | 5 471 753 | 264 | 4.8 (4.3–5.4) | 5 149 776 (94.1) | 172 (65.2) | 3.3 (2.9–3.9) | 321 977 (5.9) | 88 (33.3) | 27.3 (22.2–33.7) |
| 2015 | 5 487 308 | 272 | 5.0 (4.4–5.6) | 5 150 146 (93.9) | 151 (55.5) | 2.9 (2.5–3.4) | 337 162 (6.1) | 115 (41.9) | 34.1 (28.4–40.9) |
| 2016 | 5 503 297 | 233 | 4.2 (3.7–4.8) | 5 145 756 (93.5) | 117 (50.2) | 2.3 (1.9–2.7) | 357 541 (6.5) | 111 (47.6) | 31.0 (25.8–37.4) |
| 2017 | 5 513 130 | 246 | 4.5 (3.9–5.1) | 5 140 328 (93.2) | 139 (56.5) | 2.7 (2.3–3.2) | 372 802 (6.8) | 101 (39.0) | 27.1 (22.3–32.9) |
| Total | | 1015 | | | 579 (57.0) | | | 415 (40.3) | |

* Including cases with unknown country of birth.

**Table 3. Characteristics of TB transmission patterns in clusters including both Finnish-born and foreign-born cases in Finland, 2014–2017.**

| Spoligotype + MIRU-VNTR | No of cases | Males n | Age range, years | Pulmonary/ Smear positive | Foreign-born (%) | Foreign area of birth | Diagnosis time frame | District | Epidemiological background information | Lineages/Notes |
|---|---|---|---|---|---|---|---|---|---|---|
| SIT53 +1112–15[a] | 13 | 8 | 24–86 | 10/5 | 7,7 | Europe: 1 | 2014–2017 | Several regions | Unknown | T1 lineage, the most common in Finland |
| SIT1 +94–32[a] | 8 | 5 | 18–63 | 8/5 | 62,5 | South-Asia: 2 North-Asia: 3 | 2014–2017 | Several regions | Unknown | Beijing lineage, very common in Finland |
| SIT149 +594–15[a] | 8 | 7 | 17–63 | 8/4 | 75 | East Africa: 5 South-East Asia: 1 | 2014–2017 | Several regions | 5 cases from the same city | T3-ETH lineage, common in Finland |
| SIT42 +1119–52[a] | 8 | 5 | 30–69 | 8/6 | 25 | Europe: 2 | 2014–2015 | Several regions | 5 cases from the same city | LAM9 lineage, very common in Finland |
| SIT1 +100–32[a] | 7 | 5 | 34–78 | 6/3 | 71,4 | Europe: 3 North Asia: 2 Unknown: 1 | 2014–2017 | Regional | 6 cases from the same city | Beijing lineage, very common in Finland MDR cluster (one case is not MDR) |
| SIT1 +342–32[a] | 7 | 4 | 16–83 | 7/3 | 57,1 | East Africa: 2 South-East Asia: 2 | 2015 | Several regions | 5 cases from the same school | Beijing lineage, very common in Finland |
| SIT381 + 18194–32[a] | 7 | 5 | 13–34 | 7/3 | 42,9 | East Africa: 3 | 2015–2017 | Local | household and friendship | CAS1-Delhi lineage, not common in Finland |
| SIT50 +172–69[a] | 4 | 4 | 19–36 | 3/2 | 75 | Southwest Asia: 1 West Africa: 2 | 2015–2017 | Several regions | Unknown | Harlem3 lineage, very common in Finland |
| SIT254 + 9118–52 | 3 | 3 | 37–46 | 3/2 | 66,7 | Europe: 2 | 2014 | Several regions | Unknown | LAM-RUS lineage, not common in Finland |
| SIT2028 + 1481–66[a] | 3 | 2 | 15–61 | 3/3 | 33,3 | Europe: 1 | 2014–2015 | Local | 3 cases from the same city | Unknown lineage |
| SIT1 +3882–32 | 2 | 0 | 28–43 | 2/1 | 50 | South-East Asia: 1 | 2014–2016 | Several regions | Unknown | Beijing lineage, very common in Finland |
| SIT42 +5014–218 | 2 | 2 | 31–71 | 2/0 | 50 | East Africa: 1 | 2015 | Several regions | Unknown | LAM9 lineage, very common in Finland |
| SIT1 +3894–32[a] | 2 | 1 | 39–51 | 2/1 | 50 | South-East Asia: 1 | 2016 | Several regions | Unknown | Beijing lineage, very common in Finland |
| SIT928 + 18673–15 | 2 | 2 | 19–57 | 0/0 | 50 | East Africa: 1 | 2016 | Several regions | Unknown | T lineage, common in Finland |
| Total | 76 (2–13) | 53 (70%) | 13–86 | 69/38 | 47,4 | | 2014–2017 | | | |

[a] The cluster detected in Finland before the year 2014.

Based on the date of symptom onset or diagnosis, the first case in 7 (50%) mixed clusters was foreign-born and in 7 (50%) Finnish-born. In 13 clusters the first case had pulmonary TB, and in one cluster both cases had extrapulmonary TB. A total of 67 (88%) cases in 10 mixed clusters were

in clusters that had been detected before the year 2014 in Finland (based on the genotype of the cluster). The remaining four mixed clusters had not been detected in Finland earlier (Table 3).

Based on the epidemiological background information, in two mixed clusters the epidemiologic link was definite (same school or household/friendship) between Finnish- and foreign-born cases. In both clusters, the foreign-born case was the first case diagnosed. In four clusters, the links were less clear: one cluster included cases from the same city and in three clusters most of the cases were living in the same city. In the remaining eight clusters cases lived in several different regions in Finland. A phylogenetic tree was constructed based on MIRU-VNTR typing data to show the relatedness among mixed clusters (S1 Fig).

## Discussion

Our nationwide, population-based study showed that during the study period (2014–2017) the total number of foreign-born TB cases increased while TB incidence first increased and then decreased to its lowest level in 10 years [2]. Although a large proportion of TB cases were among foreign-born individuals, our data suggest that transmission from the foreign-born to Finnish-born population was less frequent (17.5% vs. 32% of the TB clusters were part of mixed clusters) than the median in the European Union/European Economia Area (EU/EEA) countries during 1992–2007, shown by a systematic review published in 2014 [7]. To our knowledge, recent studies of the extent of transmission including cases of the 2015–2016 major influx of asylum seekers between foreign- and native-born populations have not been published from EU/EEA countries.

Our study shows that over 50% of the clusters included isolates from Finnish-born cases only, and 30% from foreign-born cases only. Furthermore, more than half of the foreign-born clusters included cases originating only from one country. This finding is in line with other studies conducted in low-incidence countries which suggest that the risk of transmission is elevated within migrant households and migrant communities, but not between host and migrant populations [7, 8].

In a previous study conducted in Finland between 2008–2011, only 10% of clusters included both Finnish-born and foreign-born cases [12]. The increase might be due to increase in the number of foreign-born people living in Finland, and also of the assumption that Finnish-born and foreign-born are interacting more with each other [21]. It has been shown that more than half of the foreign-born TB cases are diagnosed within two years of arrival to Finland [22], which supports the suggestion that TB in foreign-born cases in Finland is often caused by a reactivation of TB obtained in their country of birth [12].

No difference was found in proportions of clustered isolates among Finnish-born and foreign-born populations as 25% of the Finnish-born cases and 24% of the foreign-born cases were part of a cluster. The clustering rate of foreign-born cases is similar in Spain, where 22–28% of the foreign-born TB cases were part of a cluster [23, 24], but other European countries have reported higher proportions; 30% in Norway [25], 35% in France [26], 46% in Italy [27] and 56% in Sweden [28]. These countries have had more immigrants for a longer period of time than Finland [29], resulting in more interaction between natives and immigrants [26] which may explain the difference.

Only 6 out of 14 mixed clusters had a definite or possible epidemiologic link between the cases. Two clusters had a definite link; one cluster was transmitted in a school and another between friends and family. In four clusters, living in the same region was the possible link. In eight clusters without a known epidemiological link the cases in the same cluster were diagnosed in several regions in Finland. It is known that migrants are very mobile leading to clusters spreading to wider geographical areas [30].

The full size of the clusters is not captured due to the fact that TB transmission chains are building up slowly, TB has a long exposure and incubation time and delays in diagnosis can be substantial. In this study, the vast majority of the mixed clusters had a genotype that had been seen in Finland earlier. The proportion of Finnish and foreign-born varied within the mixed clusters. The largest mixed cluster consisted of 13 cases, 12 of which were Finnish-born. This cluster had a genotype that is very common in Finland and has caused local outbreaks. Thus, the infection of the foreign-born case in this cluster had probably been acquired from a Finnish-born case, although this genotype is common also elsewhere. The greatest proportion of foreign-born cases were found in widely spread (MDR) clusters SIT1+100–32 and SIT1+94–32 belonging to the Beijing genotype [31].

Approximately half of the cases of the mixed clusters were Finnish-born and half foreign-born. Thus, we can assume that both Finnish-born and foreign-born cases contribute to the same extent to TB transmission in mixed clusters and the cross-transmission among foreign- and native-born populations is bidirectional in Finland, as reported also in a systematic review in the EU/EEA countries [7]. To verify this assumption, whole genome sequencing (WGS) with thorough data analysis [32] and contact tracing should be performed to get more detailed information about direction of transmission chains.

In Finland, the majority of TB cases are still reported in the Finnish-born population. Despite of the increase in number of foreign-born people, Finland has not reached the epidemiological situation reported from Sweden, Norway, Denmark and The Netherlands, where most of the TB cases are foreign-born [33–38]. It appears that the large influx of asylum seekers had a short-term influence on the number and the incidence of foreign-born TB cases. However, it did not have a major impact on the overall annual incidence of TB in Finland.

The characteristics of TB cases have remained rather stable over the years, when compared to our previous study [2]. However, some differences to the previous period can be seen: Finnish-born cases are older, foreign-born cases are younger, the proportion of cases with Somali origin has decreased and, furthermore, the frequency of smear positive pulmonary TB cases has decreased, resulting in fewer highly infectious TB cases.

There were several limitations in our study. First, while asylum seekers and other migrants, who do not have a residency permit in Finland, are waiting for a decision on the residency permit application, they are not considered permanent residents in Finland and are not registered in the population information system [39]. In 2015–2016 there were approximately 35,000 asylum seekers [38] and an unknown number of paperless people who were not registered into the population information system. For that reason, the incidence of TB among immigrants may be an overestimate as these migrants who do not have a residency permit are not counted in the population denominator in Finland. Second, we used spoligotyping and MIRU-VNTR as genotyping methods to detect the clusters. WGS method with higher discriminatory power probably would have detected less clustered cases [40, 41] or possibly could have identified more clusters by splitting clusters smaller [42]. Third, our study may underestimate transmission due to the limited time of observation. Fourth, the definitions of foreign-born was made according to the country of birth, and if not available, the most recent nationality. With the data available, we were not able to identify second-generation immigrants who have been born in Finland: this could give a clearer understanding of the transmission within the immigrant community.

## Conclusion

While both the number of immigrants and the number of TB cases among foreign-born individuals is increasing in Finland, our study suggests that transmission of TB from foreign- to

Finnish-born population is uncommon, as some 17.5% of clusters and 9.6% of the cases were part of clusters with isolates from both Finnish-born and foreign-born cases.

## Supporting information

**S1 Fig. Phylogenetic tree of the mixed clusters.** The tree was generated with Bionumerics 6.6 using the UPGMA (unweighted pair group method with arithmetic mean) method on the categorical values of the similarity matrix of the MIRU-VNTR results.
(PDF)

## Acknowledgments

We thank Jukka Ollgren for technical assistance in statistical analysis.

## Author Contributions

**Conceptualization:** Pirre Emilia Räisänen, Hanna Soini, Petri Ruutu, J. Pekka Nuorti, Outi Lyytikäinen.

**Data curation:** Pirre Emilia Räisänen, Marjo Haanperä, Hanna Soini, Outi Lyytikäinen.

**Formal analysis:** Pirre Emilia Räisänen, Hanna Soini, J. Pekka Nuorti, Outi Lyytikäinen.

**Funding acquisition:** Pirre Emilia Räisänen, Outi Lyytikäinen.

**Investigation:** Pirre Emilia Räisänen, J. Pekka Nuorti, Outi Lyytikäinen.

**Methodology:** Pirre Emilia Räisänen, Marjo Haanperä, Hanna Soini, J. Pekka Nuorti, Outi Lyytikäinen.

**Project administration:** Pirre Emilia Räisänen, J. Pekka Nuorti, Outi Lyytikäinen.

**Resources:** Pirre Emilia Räisänen, J. Pekka Nuorti, Outi Lyytikäinen.

**Software:** Pirre Emilia Räisänen, J. Pekka Nuorti, Outi Lyytikäinen.

**Supervision:** Pirre Emilia Räisänen, Marjo Haanperä, Hanna Soini, Petri Ruutu, J. Pekka Nuorti, Outi Lyytikäinen.

**Validation:** Pirre Emilia Räisänen, Petri Ruutu, J. Pekka Nuorti, Outi Lyytikäinen.

**Visualization:** Pirre Emilia Räisänen, Outi Lyytikäinen.

**Writing – original draft:** Pirre Emilia Räisänen.

**Writing – review & editing:** Pirre Emilia Räisänen, Marjo Haanperä, Hanna Soini, Petri Ruutu, J. Pekka Nuorti, Outi Lyytikäinen.

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
