## [Decision Letter · Decision Letter 0]

11 Feb 2021

PONE-D-20-38987

Transmission of tuberculosis between foreign-born and Finnish-born populations in Finland, 2014-2017

PLOS ONE

Dear Dr. Räisänen,

Thank you for submitting your manuscript to PLOS ONE. After careful consideration, we feel that it has merit but does not fully meet PLOS ONE’s publication criteria as it currently stands. Therefore, we invite you to submit a revised version of the manuscript that addresses the points raised during the review process.

The manuscript was evaluated by an expert in the field (please see his/her comments below this letter) and by myself. The reviewer (with whom I agree) raised a number concerns that should be addressed to improve clarity and presentation of your data. 

We look forward to receiving your revised manuscript.

Kind regards,

Igor Mokrousov, Ph.D., D.Sc.

Academic Editor

PLOS ONE

Journal Requirements:

2.We note that you have indicated that data from this study are available upon request. PLOS only allows data to be available upon request if there are legal or ethical restrictions on sharing data publicly. For information on unacceptable data access restrictions, please see http://journals.plos.org/plosone/s/data-availability#loc-unacceptable-data-access-restrictions.

4.We noticed you have some minor occurrence of overlapping text with the following previous publications, which needs to be addressed:

- https://www.cambridge.org/core/journals/epidemiology-and-infection/article/tuberculosis-in-immigrants-in-finland-19952013/7ED2C7F8A951802B8722AB06D552285B

- https://bmcpublichealth.biomedcentral.com/articles/10.1186/s12889-018-5501-y

In your revision ensure you cite all your sources (including your own works), and quote or rephrase any duplicated text outside the methods section. Further consideration is dependent on these concerns being addressed.

Reviewers' comments:

Reviewer's Responses to Questions

**Comments to the Author**

1. Is the manuscript technically sound, and do the data support the conclusions?

Reviewer #1: Partly

2. Has the statistical analysis been performed appropriately and rigorously? 

Reviewer #1: Yes

3. Have the authors made all data underlying the findings in their manuscript fully available?

Reviewer #1: No

4. Is the manuscript presented in an intelligible fashion and written in standard English?

Reviewer #1: No

5. Review Comments to the Author

Reviewer #1: The manuscript entitled “Transmission of tuberculosis between foreign-born and Finnish-born populations in Finland, 2014-2017” describes the epidemiology of tuberculosis and evaluates the disease transmission between foreign-born and Finnish-born populations using MIRU-VNTR. While the study attempts to understand the major concerns on TB transmission in Finland, it cannot be accepted in its current form and requires following revisions.

Minor comments:

• There are inconsistencies in writing throughout the manuscript. Mycobacterium tuberculosis is written as M. tuberculosis, MTB and Mycobacterium tuberculosis

• MTB abbreviation has not been described anywhere

• MIRU-VNTR was first used in abstract (line 25). However, the abbreviation was first described in line number 75 under methods

• In line 85 “The genotypes of the mixed clusters were 85 compared with a local genotyping database”- Please provide the link or reference to the database

• A figure describing close genetic relatedness between epidemiological risk factors (immigration status, age, history of TB) and MIRU-VNTR profiles will help in data interpretation

• Please describe line number 72- “additional information on the contact tracing (free text)”

• Under methods section, please include information on the kind of culture technique used (liquid or solid?). Considering the possibility that an individual may be infected with multiple strains, it is important to mention if the samples for DNA extraction were taken from a single or multiple colony

• Please include the DNA extraction method (kit and reagent information or a reference to the previously described method)

• Please provide information on which method was used for MIRU-VNTR typing (15 or 24-locus). Include MIRU-VNTR typing spoligotyping kit information

• It is not cleared how cluster cutoff was decided in MIRU-VNTR. For example, do you call the isolates as clusters if 24 loci matched? How do you interpret the samples with mixed MIRU-VNTR patterns?

• A phylogenetic tree could be constructed based on MIRU-VNTR typing data for better visualization of relatedness among the samples.

6. PLOS authors have the option to publish the peer review history of their article (what does this mean?). If published, this will include your full peer review and any attached files.

Reviewer #1: **Yes: **Renu Verma

---

## [Author Response · Author response to Decision Letter 0]

6 Apr 2021

Dear Editor,

Thank you for the positive response on our manuscript entitled " Transmission of tuberculosis between foreign-born and Finnish-born populations in Finland, 2014-2017" (Manuscript ID PONE-D-20-38987). We have to the best of our ability approached and answered the reviewer’s and editor’s comments and questions. We hope that our manuscript is now suitable for publication in your journal. In the revised manuscript, Figures and Tables, we have indicated changes in the text with “Track changes”.

Please find our answers and input below.

Reviewer #1: 

The manuscript entitled “Transmission of tuberculosis between foreign-born and Finnish-born populations in Finland, 2014-2017” describes the epidemiology of tuberculosis and evaluates the disease transmission between foreign-born and Finnish-born populations using MIRU-VNTR. While the study attempts to understand the major concerns on TB transmission in Finland, it cannot be accepted in its current form and requires following revisions.

Minor comments:

1. There are inconsistencies in writing throughout the manuscript. Mycobacterium tuberculosis is written as M. tuberculosis, MTB and Mycobacterium tuberculosis. MTB abbreviation has not been described anywhere

Answer: Thank you for your observation. The inconsistencies are now revised and the writing has been made consistent throughout the manuscript. MTB abbreviation has been changed to M. tuberculosis. All the changes are shown in tracked changes in the manuscript.

2. MIRU-VNTR was first used in abstract (line 25). However, the abbreviation was first described in line number 75 under methods

Answer: Thank you for your observation. In abstract, lines 27-28, MIRU-VNTR has been changes to Mycobacterial Interspersed Repetitive Unit Variable Number Tandem Repeat. 

Also a change has been made to the methods section, lines 78-80: Culture-positive M. tuberculosis isolates are routinely sent to the THL Mycobacterial Reference Laboratory for drug susceptibility testing and genotyping by spoligotyping and Mycobacterial Interspersed Repetitive Unit Variable Number Tandem Repeat (MIRU-VNTR) (24 loci) according to standard protocols [14, 15].

3. In line 85 “The genotypes of the mixed clusters were 85 compared with a local genotyping database”- Please provide the link or reference to the database

Answer:Thank for the comment. Unfortunately there is no link or reference to the local genotyping database. The data can be accessed by applying for a data permit through the Health and Social Data Permit Authority (Findata) (https://www.findata.fi/en/).

4. A figure describing close genetic relatedness between epidemiological risk factors (immigration status, age, history of TB) and MIRU-VNTR profiles will help in data interpretation

Answer: We agree with the reviewer that this kind of figure might help with the data interpretation in some circumstances. However, presenting the odds ratios for different epidemiologic risk factors graphically is not feasible because of small number (n=76) of isolates in mixed clusters in our study and would not enable drawing conclusions from the data. Whole genome sequencing (WGS) with thorough data analysis and contact tracing would be more feasible method to get detailed information about the genetic relatedness.

5. Please describe line number 72- “additional information on the contact tracing (free text)”

Answer: Thank you for the clarifying question. By free text we mean a free text column where phycisians are able to add additional information regarding the contact tracing.

Sentence is revised in the methods section, line 75: free text column on the notification form

6. Under methods section, please include information on the kind of culture technique used (liquid or solid?). Considering the possibility that an individual may be infected with multiple strains, it is important to mention if the samples for DNA extraction were taken from a single or multiple colony

Answer: Thank you for a valid point. We have included information regarding culture techniques used. 

Sentence added in the methods section, lines 81-83: DNA was extracted from bacterial colonies grown on solid medium using the CTAB (cetyl trimethylammonium bromide) method [16, 17].

7. Please include the DNA extraction method (kit and reagent information or a reference to the previously described method)

Answer: We have now included the DNA extraction method used. 

The first paragraph of the laboratory methods and identification of clusters revised, lines 79-86: Culture-positive M. tuberculosis isolates are routinely sent to the THL Mycobacterial Reference Laboratory for drug susceptibility testing and genotyping by spoligotyping and Mycobacterial Interspersed Repetitive Unit Variable Number Tandem Repeat (MIRU-VNTR) (24 loci) according to standard protocols [14, 15]. DNA was extracted from bacterial colonies grown on solid medium using the CTAB (cetyl trimethylammonium bromide) method [16, 17]. Spoligotyping was performed using spoligotyping membranes obtained from Ocinum Biosolutions (Hyderabad, P.A., India) or produced in-house as previously described by Kamerbeek et al. [15]. The MIRU-VNTR analysis was performed at GenoScreen (Lille, France) using the MIRU-VNTR Kits of the same company.

Also, two references were added and the reference list revised accordingly:

16. van Soolingen D, Hermans PW, de Haas PE, Soll DR, van Embden JD. Occurrence and stability of insertion sequences in Mycobacterium tuberculosis complex strains: evaluation of an insertion sequence-dependent DNA polymorphism as a tool in the epidemiology of tuberculosis. J Clin Microbiol. 1991;29(11):2578-2586. doi:10.1128/JCM.29.11.2578-2586.1991

17. van Beek J, Haanperä M, Smit PW, Mentula S, Soini H. Evaluation of whole genome sequencing and software tools for drug susceptibility testing of Mycobacterium tuberculosis. Clin Microbiol Infect. 2019;25(1):82-86. doi:10.1016/j.cmi.2018.03.041

8. Please provide information on which method was used for MIRU-VNTR typing (15 or 24-locus). Include MIRU-VNTR typing spoligotyping kit information

Answer: Thank you for the clarifying point. We have used 24 loci MIRU-VNTR and also kit information added. See added paragraph above.

9. It is not cleared how cluster cutoff was decided in MIRU-VNTR. For example, do you call the . isolates as clusters if 24 loci matched? How do you interpret the samples with mixed MIRU-VNTR patterns?

Answer: Thank you for this question. In our study clusters were defined based on both MIRU-VNTR and spoligotyping. Both (MIRU-VNTR and spoligotyping) needed to match so that a cluster was defined. As mentioned in the manuscript on the ”Laboratory methods and identification of clusters” section: ”When two or more M. tuberculosis isolates matched by genotyping methods (i.e., identical spoligotype and MIRU-VNTR patterns), they were considered a genotype cluster.” That said, if only MIRU-VNTR was the same with two (or more) isolates, it was not considered as a cluster.

10. A phylogenetic tree could be constructed based on MIRU-VNTR typing data for better visualization of relatedness among the samples.

Answer: A phylogenetic tree has been added as a supplementary file due to size of the tree. 

Also, a sentence added in the methods section, lines 92-94: Clustering of isolates with incomplete MIRU-VNTR profile due to missing/undetectable, double, or non-numerical (e.g. 3s) results was performed with help of UPGMA (unweighted pair group method with arithmetic mean) tree generated in the Bionumerics software.

Sentece added in the results section, lines 148-149: A phylogenetic tree was constructed based on MIRU-VNTR typing data to show the relatedness among mixed clusters (S1 Fig).

and

Answer: We have ensured that our manuscript meets PLOS ONE’s style requirements.

2.We note that you have indicated that data from this study are available upon request. PLOS only allows data to be available upon request if there are legal or ethical restrictions on sharing data publicly. For information on unacceptable data access restrictions, please see http://journals.plos.org/plosone/s/data-availability#loc-unacceptable-data-access-restrictions.

Answer: We wish we could be providing the data set, but our data includes sensitive patient information. However, the Finnish Communicable Diseases Act 1227/2016, 42 § states the following: Release of information for research The Finnish Institute for Health and Welfare may, notwithstanding confidentiality provisions, decide to release personal data from registers referred to in this Act that are maintained by the institute, if the release is made for the purpose of scientific research on health care activities, prevention or treatment of diseases, or research related to these, and if the release meets the requirements laid down in section 28 of the Act on the Openness of Government Activities (621/1999).

That said, the data permit can be applied for scientific research through the Health and Social Data Permit Authority (Findata) (https://www.findata.fi/en/)

Answer: Ethical statement was moved into the Methods section

4.We noticed you have some minor occurrence of overlapping text with the following previous publications, which needs to be addressed:

- https://www.cambridge.org/core/journals/epidemiology-and-infection/article/tuberculosis-in-immigrants-in-finland-19952013/7ED2C7F8A951802B8722AB06D552285B

- https://bmcpublichealth.biomedcentral.com/articles/10.1186/s12889-018-5501-y

In your revision ensure you cite all your sources (including your own works), and quote or rephrase any duplicated text outside the methods section. Further consideration is dependent on these concerns being addressed.

Answer: Thank you for checking out the overlapping text with previous studies. These two studies mentioned here are our own articles and the methods are same or similar. Other parts have been cited accordingly.

---

## [Editor Report · Decision Letter 1]

12 Apr 2021

Transmission of tuberculosis between foreign-born and Finnish-born populations in Finland, 2014-2017

PONE-D-20-38987R1

Dear Dr. Räisänen,

I am pleased to inform you that your manuscript has been judged scientifically suitable for publication and will be formally accepted for publication once it meets all outstanding technical requirements.

Kind regards,

Igor Mokrousov, Ph.D., D.Sc.

Academic Editor

PLOS ONE
---

## [Editor Report · Acceptance letter]

14 Apr 2021

PONE-D-20-38987R1 

Transmission of tuberculosis between foreign-born and Finnish-born populations in Finland, 2014-2017 

Dear Dr. Räisänen:

I'm pleased to inform you that your manuscript has been deemed suitable for publication in PLOS ONE. Congratulations! Your manuscript is now with our production department. 

Kind regards, 

on behalf of

Dr Igor Mokrousov 

Academic Editor

PLOS ONE